# Extending the Global Space-Based Inter-Calibration System (GSICS) to Tie Satellite Radiances to an Absolute Scale

**Tim J. Hewison** [1],*, **David R. Doelling** [2], **Constantine Lukashin** [2], **David Tobin** [3], **Viju O. John** [1], **Sauli Joro** [1] **and Bojan Bojkov** [1]

1   European Organisation for the Exploitation of Meteorological Satellites (EUMETSAT), 64295 Darmstadt, Germany; Viju.John@eumetsat.int (V.O.J.); Sauli.Joro@eumetsat.int (S.J.); Bojan.bojkov@eumetsat.int (B.B.)
2   NASA Langley Research Center, Hampton, VA 23666, USA; david.r.doelling@nasa.gov (D.R.D.); constantine.lukashin-1@nasa.gov (C.L.)
3   Space Science and Engineering Center (SSEC), University of Wisconsin, Madison, WI 53706, USA; dave.tobin@ssec.wisc.edu
*   Correspondence: tim.hewison@eumetsat.int

**Abstract:** The Global Space-based Inter-Calibration System (GSICS) routinely monitors the calibration of various channels of Earth-observing satellite instruments and generates *GSICS Corrections*, which are functions that can be applied to tie them to reference instruments. For the infrared channels of geostationary imagers GSICS algorithms are based on comparisons of collocated observations with hyperspectral reference instruments; whereas Pseudo Invariant Calibration Targets are currently used to compare the counterpart channels in the reflected solar band to multispectral reference sensors. This paper discusses how GSICS products derived from both approaches can be tied to an absolute scale using specialized satellite reference instruments with SI-traceable calibration on orbit. This would provide resilience against gaps between reference instruments and drifts in their calibration outside their overlap period and allow construction of robust and harmonized data records from multiple satellite sources to build Fundamental Climate Data Records, as well as more uniform environmental retrievals in both space and time, thus improving inter-operability.

**Keywords:** satellite instrument; inter-calibration; traceability; cross-calibration; climate data record; Earth observation

## 1. Introduction

The Global Space-based Inter-Calibration System (GSICS) is an international collaborative effort, which aims at ensuring consistent measurement accuracy among space-based observations worldwide for climate monitoring, weather forecasting, and environmental applications [1]. This is achieved through a comprehensive calibration strategy, which involves routine monitoring of instrument performances, operational inter-calibration of satellite instruments, tying the measurements to absolute references and standards, and recalibration of archived data.

One part of GSICS' strategy involves direct comparisons of collocated observations from pairs of satellite instruments, which are used to systematically generate calibration functions to compare and correct the biases of *monitored instruments* to *references*. This approach is currently applied to inter-calibrate the infrared (IR) channels of geostationary (GEO) imagers to hyperspectral sounders on Low Earth Orbit (LEO) satellites, which are used as references to generate GEO-LEO IR *GSICS Corrections*. These are derived by various satellite operating agencies from a commonly-agreed

algorithm [2], similar to the Simultaneous Nadir Overpass (SNO) method originally developed for LEO-LEO comparisons [3], but extended to include a range of view angles. This GSICS algorithm is based on the comparison of collocated simultaneous observations from pairs of satellite instruments with similar viewing geometries, using a weighted linear regression. Because a hyperspectral reference instrument is used as a calibration reference for contemporary satellites, it is possible to accurately synthesize the equivalent radiance of the multispectral monitored instrument by convolving its spectral response function with the observed scene radiance spectra. However, some reference instruments, such as CrIS and AIRS, do not provide complete or contiguous spectral coverage of all monitored instruments' IR channels. Algorithms have been developed to compensate for spectral gaps [4], which introduce additional, but quantifiable, uncertainties into the GSICS product.

An extension of the SNO approach, sometimes known as Ray-matching, could be applied to inter-calibrate counterpart channels in the reflected solar band (VIS/NIR). However, GSICS currently applies a complementary, indirect approach whereby the observations of Pseudo Invariant Calibration Targets (PICTs), such as the Moon or Deep Convective Clouds (DCCs) [5], are used to transfer the calibration of the reference instrument to the monitored instrument. These observations need not be simultaneous, but do need to be made under directly comparable conditions—for example, viewing and solar geometry. However, in the former case, it is also possible to use a lunar irradiance model to account for changes due to the Moon's phase and libration [6]. There is, however, currently no hyperspectral satellite instrument covering the full spectral band of GEO imagers' visible and near-infrared channels that would make a suitable reference. So instead, GSICS has selected S-NPP/VIIRS as a multispectral reference instrument due to its spectral characteristics, calibration stability and good quality of its characterization [7]. This necessitates the use of Spectral Band Adjustment Factors (SBAFs) [8] to account for the radiance differences introduced by the monitored and reference instruments' equivalent channels having non-identical spectral response functions.

Similar approaches have also been proposed for thermal infrared and microwave—e.g., using the Moon as a reference [9,10].

These methods are used to derive effective calibration corrections (or, equivalently, new calibration coefficients). In the GEO-LEO IR case, the current *GSICS Corrections* are defined as linear functions of the GEO radiances, based on their weighted linear regression with the reference instruments' radiances [2]. In the GEO-LEO VIS case, GSICS generates new calibration coefficients, based to convert the GEO imagers' observed counts to be consistent with radiances observed by the reference instrument over Deep Convective Clouds [5]. Both are distributed as GSICS products to facilitate interoperability and allow for accurately integrating data from multiple observing systems into operational near real-time processing, as well as for re-analysis applications.

## 2. Applying the Concept of Traceability to GSICS Products

While the concept of traceability can mean different things to different communities, when applied to GSICS products, it refers to the ability to relate the corrected radiance of the monitored satellite instrument to the community-defined reference instruments through an unbroken chain of comparisons, each with stated uncertainties. Naturally, the different levels of uncertainty associated with each comparison will affect the overall quality with which the end product is traceable back to the reference.

### 2.1. Traceability Concept Applied to Direct Inter-Calibration

For GSICS products derived by direct inter-comparison of a monitored instrument to a single reference instrument, the traceability chain is established by constructing an uncertainty budget. For example, Hewison [11] considers all processes contributing to the uncertainty on the comparisons and propagates these through a model of the comparison in a Type-B uncertainty analysis, following the *CIPM Guide to the Expression of Uncertainty in Measurement* (GUM) [12]. The GSICS GEO-LEO IR inter-calibration algorithm includes the provision of the estimated random uncertainty for each

correction by propagating noise and scene variability through the weighted regression used to generate them. These estimates can be validated using a Type-A evaluation of their time series [12].

However, it is often desirable to use multiple reference instruments—for example, to provide greater robustness, to improve diurnal coverage by using platforms with different equator crossing times, and to extend the period over which inter-calibration is possible beyond the lifetime of a single reference instrument. This allows us to ensure the full range of the monitored instrument's operating conditions are covered. In these cases, it is still possible to establish a traceability chain by selecting one reference instrument as an *Anchor Reference*. Results derived from other references are then adjusted to be consistent with those from the anchor by constructing a series of double differences between them and using these to define *delta corrections*, which are applied to each time series before they are combined. Uncertainties provided with each component *Correction* and its delta correction are used as a weighting in the composite product, which is referred to as a *Prime GSICS Correction* [13].

It is important to test the relative stability of the products derived from each reference instrument before they are combined, as the delta correction is defined over the whole overlap period between each pair of reference instruments. So continuous monitoring of their double differences is critical. However, any drift in relative difference between reference instruments before or after the overlap cannot be accounted for.

The Prime GSICS Correction approach can be applied to combine results from historical references used for Fundamental Climate Data Records (FCDRs). This was illustrated by Tabata [14], who applied the concept to recalibrate the water vapor and infrared channels of Japanese geostationary imagers using Infrared Atmospheric Sounding Interferometer (IASI), Atmospheric Infrared Sounder (AIRS), and High-resolution Infrared Radiation Sounder/2 (HIRS/2) as reference instruments.

## 2.2. GSICS Infrared Reference Uncertainty and Traceability Report

GSICS is developing a report, to support the choice of hyperspectral reference instruments (IASI, CrIS and AIRS) to inter-calibrate channels in the thermal infrared, and the use of Metop-A/IASI as the anchor reference. This represents an update of the *GSICS Traceability Statement for IASI and AIRS* [15]—with additional results, for example including CrIS [16]. This report first reviews each reference instruments' error budget and considers the traceability of their in-flight calibration to absolute (SI) scale. The report then consolidates the results from multiple in-flight comparisons of the reference instruments by different authors, including direct inter-comparison by SNO, as well as indirect comparisons by double-differencing against GEO imagers and NWP bias monitoring. The report expresses the instruments' error budgets and inter-comparisons for common sets of dates, spectral bands (both hyperspectral and broadband averages) and scene radiance (or brightness temperature). This readily allows their comparison to form a consensus on the reference instruments' relative calibration.

It is expected that similar reports will be generated in the future to support the traceability of GSICS products for other spectral bands by reviewing the calibration uncertainty of candidate reference instruments.

## 2.3. Traceability Concept Applied to Indirect Inter-Calibration

Because the use of Pseudo Invariant Calibration Targets (PICTs), including the Moon, does not require simultaneous observations from the monitored and reference instruments, a single reference instrument can be applied to any point in time—assuming the PICT itself to be stable. In this case, the reference instrument's observations of the PICT are used to characterize its reflectance over the full range of conditions for which it is to be applied—e.g., solar and viewing geometries, seasonal or lunar variations. Typically, this is performed over a period of several years, usually soon after launch, when the reference instrument's calibration is believed to be more reliable and this period defines the reference itself.

Even the best current reference sensors with channels in the visible band do not have perfectly stable calibration, due to the unaccounted optical degradation of mirrors and detectors, not monitored

by onboard calibration systems. So careful monitoring is needed and considered in the selection of this reference period. For terrestrial PICTs, there are short-term temporal drifts only because they are shorter than the sensor record. However, the drifts are unknown on decadal and longer time scales.

Traceability of the GSICS products could be established by propagating their inputs' variability and uncertainty through a measurement model representing the inter-calibration algorithm (including SBAFs) and simplified uncertainties provided with the products. This can be especially challenging where Radiative Transfer Models are used, as although it is possible to propagate uncertainties in the model's inputs, it is much more difficult to quantify the uncertainties in the model itself. Currently, VIS/NIR GSICS products are provided with uncertainty estimates based on a Type-A evaluation of their time series [5,12].

## 3. Tying GSICS Products to an Absolute Scale

Many satellite missions attempt to calibrate their instruments against SI standards before launch. GSICS, working together with CEOS, aims to define best practices to characterize satellite instruments pre-launch using SI-traceable references to tie them to an absolute radiance scale. However, this traceability chain may be compromised during launch, due to uncontrolled changes to the instrument and its operating environment. For this reason, GSICS endorses the establishment of an observing system with calibration directly traceable to SI-standards on-orbit to act as an inter-calibration reference, which could be used to anchor inter-calibration products to an absolute scale—a long-term aim of GSICS.

A satellite mission, such as CLARREO Pathfinder with HyperSpectral Imager for Climate Science (HySICS) [17], TRUTHS [18], in which an instrument is launched, whose SI-traceability is verifiable in orbit could be used as a reference to achieve this goal. These are generically referred to here as SI-Traceable Satellite instruments (*SITSATs*).

In the simplest case, shown by the green arrow in Figure 1a, such a SITSAT can be used as a reference in direct inter-calibration of the *monitored instrument* by SNO or Ray-matching instead of the current reference instrument(s) for the IR case, or the PICTs in the VIS/NIR case. However, the coverage of the available collocated observations can be limited, depending on the design of the SITSAT—and the duration of such inter-calibration products is limited to its operating lifetime.

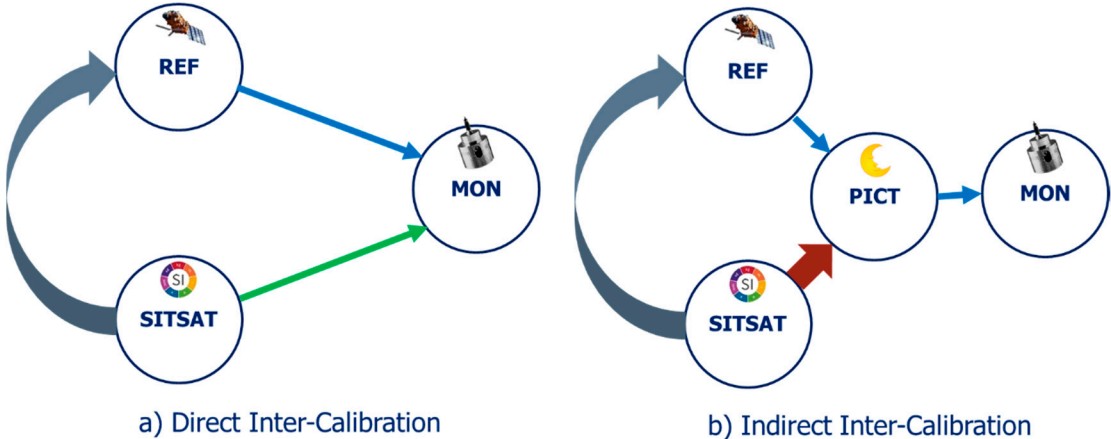

**Figure 1.** Different approaches to provide traceability of inter-calibration products for a *Monitored instrument* (MON) to an SI-Traceable Satellite instrument (SITSAT); Left (**a**): Direct Inter-Calibration: Using Ray-matching/SNO-like approach to transfer the calibration of Reference Instrument (REF) or SITSAT to MON (blue/green arrows); Right (**b**): Indirect Inter-Calibration: Using a Pseudo Invariant Calibration Target (PICT) to transfer calibration of REF or SITSAT to MON (blue/red arrows). Alternatively, the SITSAT calibration can be transferred to MON by first recalibrating the REF—either directly or indirectly, as shown by the curled grey arrows in (**a**) and (**b**).

Currently, GSICS products take the reference instruments as the truth—no uncertainty is associated with their calibration. To compare the radiances after applying the GSICS correction to an absolute scale requires the reference instruments' uncertainty to be included—and this typically dominates their overall uncertainty budget. This situation would change if it were possible to tie the inter-calibration to an absolute scale through the use of an SI-Traceable Satellite Instrument (SITSAT). SITSAT designs typically have a better calibration accuracy than the current reference instruments used in GSICS. Furthermore, SITSAT sensors can be designed with pointing abilities, which would significantly reduce the angular matching uncertainty compared to current cross-track scanning reference sensors. Therefore, their introduction could reduce the overall uncertainty with which inter-calibration can be applied to an absolute scale, as shown by examples in Table 1.

**Table 1.** Example Standard Uncertainties (coverage factor, *k*=1) in the calibration for Geostationary Imagers in different spectral bands—before and after calibration correction, shown as required and typical calibration accuracy and typical uncertainty associated with the GSICS Calibration/Correction algorithm, the current inter-calibration reference and those a SI-Traceable Satellite instrument used as a reference. Systematic uncertainty of reference instrument's calibration needs to be added to that of the GSICS Correction to obtain the combined uncertainty.

| Uncertainty Contribution/Requirement | VIS (0.4–0.75 μm) | NIR (0.75–1.3 μm) | SWIR (1.3–3.0 μm) | TIR (3–15 μm) |
|---|---|---|---|---|
| 2014 GEO Imagers Typical Calibration Accuracy | ~7% [19,20] ~4% [21,22] | ~ 7% [19] ~4% [1] | ≲ 5% [19] ~5% [1] | ≲1K [21,23] |
| 2020 GEO imagers Typical Calibration Accuracy | ~3% [24] <5% [25] | ~3% [24] <3% [25] | ~3% [24] ≲5% [25] | ≲0.2K [24] ≲0.2K [25] |
| 2022 GEO imagers Required Calibration Accuracy | <5% [26] <3% [27] | <5% [26] <3% [27] | <5% [26] <4% [27] | <0.7K [26] <1.0K [27] |
| GSICS Correction Method (excl. Reference contributions) | <1% [5] | ~1% [28] | ~3% [28] | ≲ 0.1K [11] |
| GSICS Reference Calibration Accuracy | | ≲2% [29] | | ≲ 0.1K [16,30] |
| GSICS Correction using SITSAT as Reference | | 0.15% [31] | | 0.02K [32] |
| SITSAT Reference Calibration Accuracy | | 0.3% [33] | | 0.03K [32] |

In Table 1, a) *GSICS Correction Method* is an estimate of the uncertainty on the corrected radiances introduced by the inter-calibration algorithm—e.g., [11]. It assumes a single reference instrument as the truth and does not associate an uncertainty with it. b) *GSICS Reference Calibration* represents the calibration accuracy of the reference. In the case of the Prime GSICS Correction, a) would include the uncertainty introduced by the delta corrections, propagated through the blending process, and b) would be the calibration accuracy of the Anchor reference.

Table 1 also includes some typical values for the required calibration accuracy for current and near-future geostationary imagers. These values are necessarily oversimplified in a table of this nature, but serve as an illustration. Ideally, these requirements should be technology-free, but will vary by application. In case an instrument's actual performance is outside the requirements, GSICS Corrections could be applied to bring it back within desired limits. Presenting the capabilities for the current observing system, and using SITSATs allows users of different applications to judge whether they are useful now—or potentially useful in future.

### 3.1. Tying GSICS Infrared Products to an Absolute Scale

In another approach, for direct inter-calibration methods, as used in current GSICS GEO-LEO IR products, the hyperspectral reference instrument itself can be inter-calibrated with a SITSAT (curled arrow in Figure 1a). For example, the SNO method can be applied to compare it with a sun-synchronous reference instrument's calibration with a *k*=3 uncertainty <0.1K within 2 months

of collocations, which would span a range of latitudes [32]. A correction would then be derived to transfer the reference instrument to the on-orbit SI-standard, which would be applied in addition to the current GSICS product.

This approach can be combined with the Prime GSICS Correction to extend support to FCDR generation, based on a series of double-differences to tie the whole time series to the SITSAT reference at one point in time, following the principle demonstrated in [14,34].

*3.2. Tying GSICS Reflected Solar Band Products to an Absolute Scale*

The same approach can be applied to inter-calibrate reference sensors used for indirect inter-calibration methods, such as used in the current GSICS GEO-LEO VIS/NIR products, using a SITSAT with hyperspectral VIS/NIR bands. This would then be transferred to GSICS products derived either from PICTs or direct comparisons with current reference instruments, as shown by the blue arrows in Figure 1b. However, because of the additional constraints needed to align solar, as well as viewing geometries, the number of comparisons is more limited. Roithmayr et al., [31] suggest a steerable SITSAT would be able to meet requirements for on-orbit direct inter-calibration of VIIRS' reflected solar band channels' with a $k=2$ uncertainty of 0.3% within 1 year. However, this capability depends on the polarization sensitivity of the monitored instrument, which may necessitate additional constraints on the inter-calibration sampling [35].

Another approach can be considered for indirect inter-calibration methods, in which the PICTs themselves are characterized by the SITSAT, as shown by the red arrow in Figure 1b. If these observations covered the full range of viewing conditions, it would be possible to transfer the SITSAT calibration via the PICT. The particular challenges for terrestrial targets include consistent PICT identification to ensure that the atmospheric, aerosol and residual cloud column is observed in a standard way, such that their contributions to the comparison's uncertainty do not compromise its traceability. Further analysis suggests that operating CLARREO Pathfinder on the International Space Station would yield approximately 30 samples/year of terrestrial targets, such as Libya-4 [35]. As these observations are near-nadir, it may be difficult to use this approach to fully characterize the site—although it may be possible to tie BRDF models constructed using SITSAT referenced sensors to such observations. However, such a SITSAT could observe the Moon with good coverage of the phase/libration cycle within 1 year [T. Stone, 2020—submitted for this Special Issue of *Remote Sensing*] and these observations could tie an existing model to the absolute scale—although a longer period would be needed to derive an entirely new lunar irradiance model.

A final option would be to use the SITSAT as a reference to directly inter-calibrate the GEO or LEO imagers, as shown by the green arrow in Figure 1a—although that is obviously only applicable to the operational lifetime of the SITSAT.

However, any of these approaches to tie GSICS products for VIS/NIR channels to SITSAT would benefit from careful coordination with its operators to ensure its observing strategy provides sufficient sampling to achieve the required uncertainty in the inter-calibration product. A tool, such as [36], could be valuable for planning these observations. In practice, GSICS will investigate a multi-method approach, combining the direct inter-calibration of the reference instrument, and the characterization of multiple PICTs. The balance between these approaches will depend on specific needs for each PICT and the capabilities of the SITSAT, and will evolve as different SITSATs become available.

## 4. Application of GSICS Products for Consistent Cross-Sensor Retrievals

The goal of GSICS is to produce sensor-specific calibration datasets for enabling long-term, consistent GEO and LEO cloud, aerosol, land use, and other environmental retrievals. It is important to understand the specific capabilities of GSICS in this overall effort. In an ideal scenario, consistent retrievals would be achieved by having multiple exact copies of a sensor that have perfect onboard calibration systems. Given the reality of the current constellation of sensors, however, there are many considerations that must be weighed, which are summarized by Figure 2.

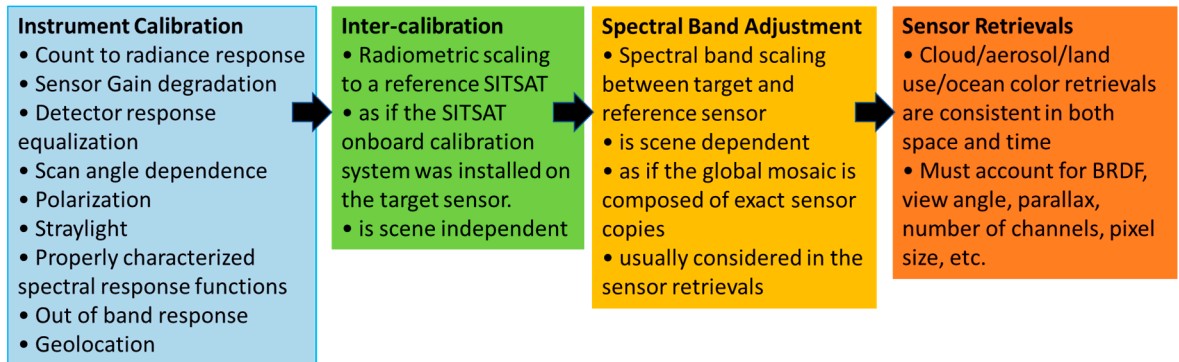

**Figure 2.** The calibration and retrieval sequence needed to radiometrically scale an instrument's signal to an SI-traceable reference to achieve consistent cross-platform retrievals.

The dedicated agency instrument calibration teams are best suited to resolve the sensor dependencies as described in the blue box (Figure 2). Here, SITSAT measurements can be used for such identification and mitigation. Ideally, the onboard calibration issues are resolved by the calibration teams, with corrections made available (or applied) in the L1 product before radiometrically scaling the monitored instrument's calibration to that of the SITSAT, otherwise unresolved calibration issues might bias the subsequent steps. Next in the sequence is the *Inter-calibration* step (Figure 2 green box), in which the radiometric scaling is applied to the level L1 dataset. GSICS calibration products are specifically tied to the sensor L1 version relative to the reference calibration, or SITSAT. The GSICS calibration products provide the scaling needed to adjust the observation provided by the monitored instruments' operational calibration system to the SITSAT absolute reference. If a narrowband sensor, such as VIIRS, is used as the reference, then the radiometric transfer is based on the monitored instrument's SRF. It must be noted that the scaling does not yield equal radiance/reflectance values between the monitored and reference instruments if their SRFs differ because Earth-viewing spectra is dependent on scene type (surface classification, cloud properties, and atmospheric column).

Consistent calibration among sensors is only one of the many factors (Figure 2 yellow and orange boxes) that need to be considered in order to provide uniform sensor retrievals. For example, cloud masking requires confident pixel-level determination of clear-sky or overcast conditions, and therefore pixel resolution can significantly influence the cloud properties perceived by different sensors. In this case, the spectral band adjustments need to be scene dependent, which requires proper scene identification as best determined in the retrieval process; for example, using the SITSAT spectra in the retrieval process. These same SBAFs are used to remove dependencies owed to spectral band radiance difference when characterizing invariant ground sites in order to transfer the calibration from the target sensor to VIIRS.

*Example: Impact of GSICS Corrections for Infrared Channels of Meteosat/SEVIRI on L2 Products*

Here we illustrate the impact GSICS Corrections can have on Level-2 products derived from the infrared channels of geostationary imagers (green box in Figure 2), and contrast these with the impact of other important factors to be considered (in the orange box in Figure 2).

After EUMETSAT relocated Meteosat-8 to 41.5°E to provide Indian Ocean Data Coverage (IODC) service in 2016, it was found that the mid-level relative humidity in the Tropospheric Humidity product (THU) [37], had a time varying bias with respect to the counterpart product generated from Meteosat-10 at 0°E in their respective overlap areas. Subsequent investigation found the 7.3 µm water vapour channel (used to derive this parameter) on Meteosat-8 to have developed a small bias with respect to IASI. Although still within the nominal requirements for absolute calibration, this difference was found to be sufficient to explain the observed difference in THU. Furthermore, analysis of an initial case study confirmed the application of the alternative calibration coefficients based on the

GSICS Correction reduced the difference in THU between Meteosat-8 and -10 from -1.8% to -0.1%, as illustrated in Figure 3.

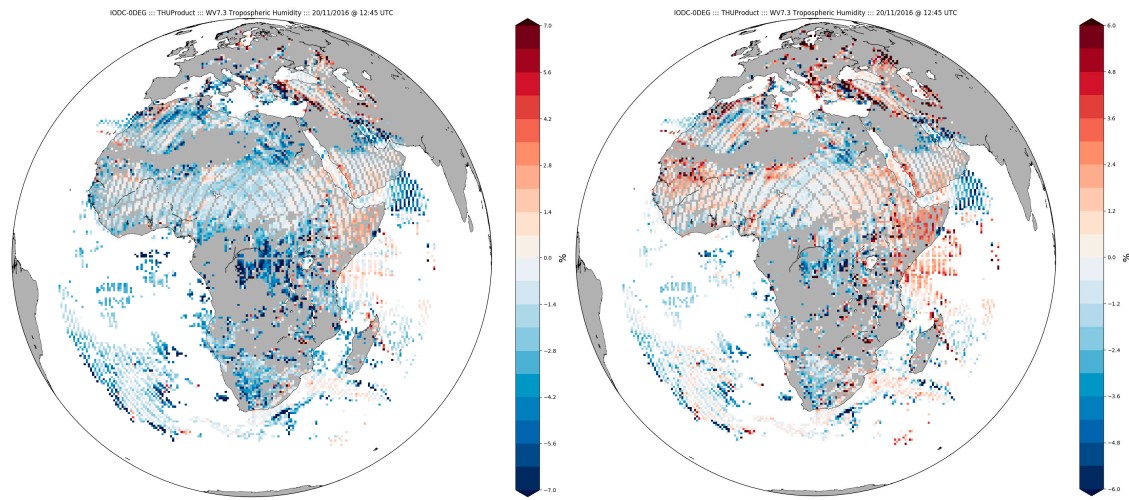

**Figure 3.** Difference of Tropospheric Humidity (THU) product derived from SEVIRI's WV7.3 channels of Meteosat-8—Meteosat-10 at 12:45Z on 20 Nov 2016 before GSICS Correction (**left**) and after GSICS Correction (**right**).

However, further analysis revealed the relative difference in the THU product derived from these instruments to vary during the day by ~1–2%. The source of this difference was eventually traced to inaccuracies in the sub-satellite position. The actual location of the satellite was only considered in the processing every 6-hours throughout the day. Other times, the satellite position was retrieved through interpolation between two known positions in the radiative transfer calculations. Due to the high inclination of Metosat-8's orbit, ±5.6° N/S/24h at the time of the investigation, the selected processing approach introduced errors of up to 2° in the calculated View Zenith Angle, which is planned to be corrected in the operational processing in 2020. This is equivalent to a ±0.3 K change in the Clear Sky Radiance at mid-latitudes in the 7.3 μm channel, which is of the same order as the calibration changes in this channel.

The application of the GSICS corrections was also found to have a small impact on other Level-2 products utilizing SEVIRI infrared channels [38]. A second case study period of 1-15 June 2018 compared Level-2 products using GSICS calibration to products using the operational calibration. An analysis of this trial suggests the GSICS correction reduces the mean differences between these products derived from Meteosat-11 and Meteosat-8 as follows: Total Precipitable Water Vapour by 0.02 mm, Tropospheric Humidity (7.3 μm channel) by 0.4%, Cloud Optical Thickness by 0.22 and, Cloud Top Pressure by 10.3 hPa. However, this analysis assumes these products to have comparable atmospheric states, on average, over the areas observed by each satellite.

While these case studies illustrate the potential benefits of applying the GSICS corrections in the operational processing chain of higher order geophysical products to improve inter-operability, they also highlight the importance of correctly handling all aspects of an instrument's observation state vector.

## 5. Discussion and Conclusions

The benefits of being able to inter-calibrate satellite instruments to an absolute scale include the resilience against gaps between reference instruments and drifts in their calibration outside their overlap period. This would allow construction of robust and harmonized data records from multiple satellite sources to build Fundamental Climate Data Records, as well as more uniform environmental retrievals in both space and time, thus improving inter-operability.

Additional benefits may be realized depending on the design of the SI-traceable reference instrument and its operating platform. In particular, hyperspectral instruments covering the full spectral band of popular channels in the visible, near-infrared and thermal infrared could accurately simulate their radiances of multi-spectral monitored instruments, making an ideal reference instrument. Such hyperspectral instruments could also be used to validate any SBAFs or similar algorithms to compensate for spectral gaps in other reference instruments, as well as characterize the spectral characteristics of various PICTs. Furthermore, by covering the full dynamic range, it could be possible to resolve other instruments' nonlinearity. Furthermore, by operating such an instrument on a non-sun-synchronous platform, any diurnal variations in the monitored instrument's calibration could be accounted for, multiple reference instruments could be inter-calibrated, and terrestrial PICTS could be characterized over the full range of solar geometries. Finally, because SITSAT designs typically have better calibration accuracy than the current reference instruments used in GSICS, they could reduce the overall uncertainty with which inter-calibration can be applied to an absolute scale, as shown in Table 1.

However, a number of challenges remain for GSICS. Primarily, the approaches described above need to be refined and applied to other inter-calibration methods. In particular, careful consideration needs to be made to how different SITSAT observing strategies could be exploited to monitor the degradation of sensor optical components not resolved by onboard calibration systems—for example, scan angle and polarization dependence introduced by the scanning mirrors. This also requires that GSICS priorities which PICT targets to characterize. Thus, to optimize the benefits of such a SI-traceable reference requires cooperation between GSICS and its operators to ensure sufficient acquisitions are available.

GSICS products have already been shown to support inter-operability in Level-2 processing chains. However, ultimately, the ability to inter-calibrate satellite instrument to an SI-standard provides irrefutability of scientific observations. Therefore, a priority for the satellite calibration community should be to establish suitable reference satellite instruments with SI-traceable calibration on-orbit. While there would still be a need for continuous monitoring to validate the instruments' calibration and ensure consistency, multiple SITSATs may eventually provide sufficient spectral, geometric and temporal coverage and long-term continuity to replace the role of current reference instruments used in GSICS inter-calibration products.

**Author Contributions:** Conceptualization, T.J.H., D.R.D.; methodology, D.T., C.L.; formal analysis, T.J.H, S.J.; supervision, B.B., writing—original draft preparation, T.J.H.; writing—review and editing, D.R.D, C.L., V.O.J., S.J., B.B.; visualization, T.J.H., D.R.D. All authors have read and agreed to the published version of the manuscript.

**Funding:** This research received no external funding.

**Conflicts of Interest:** The authors declare no conflict of interest.

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
