# Peer review of "Extending the Global Space-Based Inter-Calibration System (GSICS) to Tie Satellite Radiances to an Absolute Scale"

_remotesensing, doi:10.3390/rs12111782_

Round 1

Reviewer 1 Report

I recommend this paper for publication in Remote Sensing journal. The paper describes the Global Space-based Inter-Calibration System (GSICS) a way to construct robust and harmonized data set from raw records from satellite instruments.

Author Response

Thank you for the concise review.

Reviewer 2 Report

Thank the authors for giving this new SI-traceable concept. Overall, this is a very good paper. Just a few minor comments,

  1. a) and b) are missing in Figure 1. The authors seem to forget discussing Figure 1b.
  2. The explanations and full names of the abbreviations should be given first, e.g., Typ. Cal. Acc. Req. excl. contr., k, in table 1.
  3. It would be better if the authors could give more details on GSCIS correction for people who are not familiar with that, such as what are type-A, delta correction ….
  4. Typo in Figure 2, ‘throughput’?
  5. Please change a high-resolution image for Figure 3.
  6. I do agree with the authors that the GSICS correction approaches described need to be refined step by step in the near future.

Author Response

Thank you for the review and spotting these details, which we have addressed in our revision as described below. We appreciate this help as it has improved the manuscript.

  1. Added a) and b) to Figure 1.
    Figure 1b) is discussed later in the paper - at lines 219 and 226. We have added explicit references to Figure 1b there now.
  2. We have spelled out all the abbreviations used in Table 1.
  3. Thank you for pointing this out. We have added further explanation of the GSICS products in the introduction (lines 68-72).
    We feel the delta corrections are adequately introduced in section 2.1.
    We have revised the terminology slightly to refer to the Type-A evaluation and added further references to the GUM, which describes in more detail this standard approach of estimating the uncertainty from the variance of repeated measurements. 
  4. We revised the Figure 2 to change “sensor throughput” to the more conventional “sensor gain”.
  5. We have replaced Figure 3 with high-resolution versions of the images. 
  6. Thank you. This is a topic for further development within the GSICS Research Working Group.

Reviewer 3 Report

The authors provide a complete introduction, and they clearly explain how to apply the traceability concept to GSICS products. In addition, they present they clearly present their approach to tie GSICS products to an absolute scale.  They also provide a workflow that can achieve consistent cross platforms retrievals and present an example of how this workflow can be implemented and improve the measurements of currently deployed satellites.

Author Response

Thank you for the clear and concise review.